# Systemic Inflammation Response Index (SIRI) and Monocyte-to-Lymphocyte Ratio (MLR) Are Predictors of Good Outcomes in Surgical Treatment of Periprosthetic Joint Infections of Lower Limbs: A Single-Center Retrospective Analysis

**DOI:** 10.3390/healthcare12090867

**Published:** 2024-04-23

**Authors:** Raffaele Vitiello, Alessandro Smimmo, Elena Matteini, Giulia Micheli, Massimo Fantoni, Antonio Ziranu, Giulio Maccauro, Francesco Taccari

**Affiliations:** 1Dipartimento di Ortopedia, Fondazione Policlinico Universitario Agostino Gemelli IRCCS, 00168 Rome, Italy; raffaele.vitiello@policlinicogemelli.it (R.V.); antonio.ziranu@policlinicogemelli.it (A.Z.); giulio.maccauro@policlinicogemelli.it (G.M.); 2Department of Orthopedic and Traumatology, Villa Stuart Sport Clinic-FIFA Medical Centre of Excellence, 00135 Rome, Italy; alessandro.smimmo@gmail.com; 3Dipartimento di Sicurezza e Bioetica—Sezione di Malattie Infettive, Università Cattolica del Sacro Cuore, 00168 Rome, Italy; 4Dipartimento di Scienze di Laboratorio ed Infettivologiche, Fondazione Policlinico Universitario Agostino Gemelli IRCCS, 00168 Rome, Italy; taccari@hotmail.it

**Keywords:** prosthetic joint infection, two-stage revision, inflammatory blood markers, SIRI, MLR, delta-SIRI, delta-PCR

## Abstract

Background: Periprosthetic joint infection (PJI) is a devastating complication that develops after total joint arthroplasty (TJA), whose incidence is expected to increase over the years. Traditionally, surgical treatment of PJI has been based on algorithms, where early infections are preferably treated with debridement, antibiotics, and implant retention (DAIR) and late infections with two-stage revision surgery. Two-stage revision is considered the “gold standard” for treatment of chronic prosthetic joint infection (PJI) as it enables local delivery of antibiotics, maintenance of limb-length and mobility, and easier reimplantation. Many studies have attempted to identify potential predicting factors for early diagnosis of PJI, but its management remains challenging. In this observational retrospective study, we investigated the potential role of inflammatory blood markers (neutrophil-to-lymphocyte ratio (NLR), monocyte-to-lymphocyte ratio (MLR), platelet-to-lymphocyte ratio (PLR), systemic inflammatory index (SII), systemic inflammatory response index (SIRI), and aggregate index of systemic inflammation (AISI)) as prognostic factors in two-stage exchange arthroplasty for PJI. Methods: A single-center retrospective analysis was conducted, collecting clinical data and laboratory parameters from patients submitted to prosthetic explantation (EP) for chronic PJI. Laboratory parameters (PCR, NLR, MLR, PLR, SIRI, SII, and AISI) were evaluated at the explantation time; at 4, 6, and 8 weeks after surgery; and at reimplantation time. The correlation between laboratory parameters and surgery success was evaluated and defined as infection absence/resolution at the last follow-up. Results: A total of 57 patients with PJI were evaluated (62% males; average age 70 years, SD 12.14). Fifty-three patients with chronic PJI were included. Nine patients underwent DAIR revision surgery and chronic suppressive therapy; two patients died. Nineteen patients completed the two-stage revision process (prosthetic removal, spacer placement, and subsequent replanting). Among them, none showed signs of reinfection or persistence of infection at the last available follow-up. The other twenty-three patients did not replant due to persistent infection: among them, some (the most) underwent spacer retention; others (fewer in number) were submitted to resection arthroplasty and arthrodesis (Girdlestone technique) or chronic suppressive antibiotic therapy; the remaining were, over time, lost to follow-up. Of the patients who concluded the two-stage revision, the ones with high SIRI values (mean 3.08 SD 1.7 and *p*-value 0.04) and MLR values (mean 0.4 SD 0.2 and *p*-value 0.02) at the explantation time were associated with a higher probability of infection resolution. Moreover, higher variation in the SIRI and PCR, also defined, respectively, as delta-SIRI (mean −2.3 SD 1.8 and *p*-value 0.03) and delta-PCR (mean −46 SD 35.7 and *p*-value 0.03), were associated with favorable outcomes. Conclusions: The results of our study suggest that, in patients with PJI undergoing EP, the SIRI and MLR values and delta-SIRI and delta-PCR values could be predictive of a favorable outcome. The evaluation of these laboratory indices, especially their determination at 4 weeks after removal, could therefore help to determine which patients could be successfully replanted and to identify the best time to replant. More studies analyzing a wider cohort of patients with chronic PJI are needed to validate the promising results of this study.

## 1. Introduction

Periprosthetic joint infection (PJI) is defined as infection involving the joint prosthesis and adjacent tissue [1]. It is a devastating complication that develops after total joint arthroplasty (TJA) with an incidence of 1–3% following primary TJA and 3–5% following revision TJA [2,3,4,5,6], thus leading to a high burden for individual patients as well as the global health care systems [7]. In fact, PJIs are associated with higher hospital costs, longer hospital stays, and even greater disability and mortality [8]. Due to the growing number of TJA implanted and the comorbidity burden of the patients treated, the incidences of PJIs are expected to increase in the recent future [9,10].

Consequently, orthopedic research focused on PJI increased constantly in the last years, from 30 articles in 2008 to 364 in 2017 [11].

Traditionally, the surgical treatment of PJI has been based on algorithms where early infections are preferably treated with debridement, antibiotics, and implant retention (DAIR) or the more recent debridement, antibiotic pearls, and retention of the implant (DAPRI), while late infections are treated with two-stage revision surgery [12,13]. However, spacer retention after the first stage of an intended two-stage revision has become a viable treatment option (a so-called “1.5-stage exchange arthroplasty”), as both clinical outcomes and reinfection rates have also been reported to be acceptable with chronic infections [14,15]. Nevertheless, two-stage revision arthroplasty remains the “gold standard” for chronic PJI in North America and East Asia [16,17], with success rates ranging from 65 to 100% [18].

Many studies have attempted to identify potential predicting factors for the early diagnosis of PJI, but its management remains challenging. In works recently conducted, researchers introduced multiple measurable blood biomarkers as potential predictors of the response rate, but no definitive relationship has been found so far. Moreover, improvements in diagnostic tools, mainly to obtain more accurate diagnostic information from blood tests, are necessary [19].

In 2021, the European Bone and Joint Infection Society (EBJIS) published its latest criteria for the diagnosis of PJI [20]. According to this last revision, the cut-offs and ranges of the biomarkers were revised and new techniques like sonification and nuclear radiology were included. To confirm the presence of a prosthetic infection, one of the following criteria must be present: a sinus tract; >3000/μL leukocytes or >80% PMN in the synovial fluid; increased synovial fluid alpha-defensin (positive immunoassay or lateral-flow assay); positivity of ≥2 samples with the same pathogen; >50 CFU/mL of any organism after sonication; and the presence of ≥5 neutrophils in ≥5 one high-power field HPF or the occurrence of visible microorganisms in the histology.

Considering the complex combination of parameters necessary for the definition of PJI and the current constant need for additional diagnostic tests, many authors [21,22] underlined the importance of using all the available instruments to approach PJI diagnosis, considering the low negative predictive value of the current diagnostic methods [23]. In addition, many studies have recently emphasized the urgency of identifying new tools and parameters that can be considered complementary to the well-established diagnostic approaches for PJI. Specifically, it has been pointed out how the systemic inflammatory status and infectious processes not only determine the mere numerical value of leukocytes and platelets but more precisely they influence the relationships between these cells, for instance, the monocyte-to-lymphocyte ratio (MLR), neutrophil-to-lymphocyte ratio (NLR), and platelet-to-lymphocyte ratio (PLR) [24,25,26,27].

In this retrospective study, we investigated the role of inflammatory blood markers (neutrophil-to-lymphocyte ratio (NLR), monocyte-to-lymphocyte ratio (MLR), platelet-to-lymphocyte ratio (PLR), systemic inflammatory index (SII), systemic inflammatory response index (SIRI), and aggregate index of systemic inflammation (AISI)) in two-stage exchange arthroplasty for chronic PJI. We wanted to determine the diagnostic performance of any of them as a prognostic factor in predicting those patients who may be candidates for successful prosthesis reimplantation. To the best of our knowledge, no studies have evaluated this correlation.

## 2. Materials and Methods

All the patients admitted to the Orthopedic Department of our Institution from 1 January 2020 to 31 December 2022, affected by PJI, were retrospectively reviewed. A retrospective observational study following the PROCESS (Preferred Reporting Of CasE Series in Surgery) guidelines was carried out [28].

A single-center observational retrospective analysis was conducted, collecting clinical data and laboratory parameters from those patients submitted to prosthetic explantation for chronic PJI in a real-world setting. The data were registered from January 2020 to December 2022. All the patients with a diagnosis of PJI at our institution are routinely managed by the dedicated Bone and Joint Infection Unit. All measures taken were in accordance with the Declaration of Helsinki of 1964 and its subsequent amendments. All the procedures performed were in accordance with the 1964 Helsinki declaration and its later amendments. All enrolled patients gave informed consent to participate in the study and to the processing of their clinical data.

Forty-two patients who were undergoing revision surgery because of PJI were individuated. Among them, we performed a retrospective analysis of prospectively collected data and we identified that nineteen patients had completed the two-stage revision. Twelve patients were affected by hypertension, five patients were affected by diabetes, and two patients had no chronic pathology.

The inclusion criteria were patients with hip or knee arthroplasty with a clinical and laboratory diagnosis of confirmed PJI according to the latest criteria proposed by EBJIS in 2021 [20], who adhere to a two-stage revision strategy. The exclusion criterion was patients lost to follow-up. Age was not considered an exclusion criterion. All the patients included underwent prosthetic explantation according to the diagnosis. All the procedures were performed by orthopedic surgeons trained in bone infection treatment. The outpatient follow-up was performed by an Infectious Disease Specialist and an orthopedic surgeon with deep expertise in bone and joint infection management.

According to the definition stated by the European Bone and Joint Infection Society (EBJIS) guidelines [20], to set the diagnosis of PJI, one of the following criteria must be met: the presence of a sinus tract; >3000/μL leukocytes or >80% PMN in the synovial fluid; increased synovial fluid alpha-defensin (positive immunoassay or lateral-flow assay); positivity of ≥2 samples with the same pathogen; >50 CFU/mL of any organism after sonication; and the presence of ≥5 neutrophils in ≥5 one high-power field HPF or occurrence of visible microorganisms in histology. The newest EBJIS criteria emerged as the most sensitive of all the major definitions in ruling out infection preoperatively [29]. “*Chronic PJI*” was defined as the infection that, regardless of origin, presented in a delayed form (≥4 weeks after surgery (usually 3 months to 3 years)) associated with at least 3 weeks of symptom duration [20].

In our hospital, the two-stage revision strategy consists of the first stage, in which the prosthetic implant is removed, multiple microbiological samples are collected, an antibiotic-loaded spacer is implanted, and antibiotic therapy is started. The typical time before replanting is 8–10 weeks; antibiotic therapy is still ongoing at the time of replanting. The second stage is performed when there are no clinical signs of infection, and the C-reactive protein is stably normalized. In the second stage, the spacer is removed, multiple microbiological samples are collected, and a new implant is placed. Antibiotic therapy is discontinued upon acquisition of negative intraoperative microbiological specimens.

Blood samples of the patients were collected at the explantation time; at 4, 6, and 8 weeks after surgery; and at reimplantation time. The blood samples were examined and the inflammatory markers such as PCR, NLR, PLR, MLR, SIRI, AISI, and SII were calculated. Moreover, the main variation (laboratory parameters at the time of PJI diagnosis to laboratory parameters at reimplantation time) in these parameters was calculated and it was identified as delta-PCR, delta NLR, delta PLR, delta MLR, delta-SIRI, delta AISI, and delta-SII (Figure 1).

Treatment success was achieved if no clinical or microbiological signs of infection were assessed at the last follow-up.

GraphPad QuickCalcs (v. 2024) (GraphPad Software, San Diego, CA, USA) was used for the data analysis. The data were reported as the mean and standard deviation (±SD). A paired *t*-test was performed to compare the values pre-surgery and post-surgery. An un-paired *t*-test was used to compare the patients who completed the two-stage revision surgery. The significance was set for *p* ≤ 0.05.

## 3. Results

A total of 57 patients with PJI were evaluated (62% males; average age 70 years, SD 12.14). Fifty-three patients with chronic PJI were included. Nine patients underwent DAIR revision surgery and chronic suppressive therapy; two patients died. Nineteen patients completed the two-stage revision process (prosthetic removal, spacer placement, and subsequent replanting). Among them, none showed signs of reinfection or persistence of infection at the last available follow-up of a minimum 1 year after surgery. The other twenty-three patients did not replant due to persistent infection. The demographic features and number of involved infection sites are shown in Figure 2.

Eleven patients with chronic PJI were excluded: nine of them underwent DAIR revision surgery and chronic suppressive therapy; the other two patients died.

The average follow-up was 11 months (IQR 3–19).

The patients who showed at the time of prosthetic removal higher SIRI values (mean 3.08 SD 1.7 and *p*-value 0.04) and higher MLR values (mean 0.4 SD 0.2 and *p*-value 0.02) had a higher success rate of the EP.

Moreover, higher values of delta-SIRI (mean −2.3 SD 1.8 and *p*-value 0.03) and delta-PCR (mean −46 SD 35.7 and *p*-value 0.03) were associated with a favorable outcome. Examining the results collected, the values of the SIRI and PCR at 4 weeks from the EP were higher in the patients who showed a successful result at reimplantation and during the follow-up. The trend of the other examined parameters did not demonstrate a statistically significant correlation with the outcome of prosthesis explantation or with the two-stage revision final success.

The SIRI and PCR trends from the explantation to 4 weeks from it are shown in Figure 3.

## 4. Discussion

The diagnosis of PJI depends on the combination of clinical symptoms and laboratory and microbiological results. The most recent PJI definition has been recently proposed by the European Bone and Joint Infection Society (EBJIS) in 2021 [20]. According to these last criteria for the diagnosis of PJI, the cut-offs and ranges of the biomarkers were revised and new techniques like sonification and nuclear radiology were included. To confirm the presence of a prosthetic infection, one of the following criteria must be present: a sinus tract; >3000/μL leukocytes or >80% PMN in the synovial fluid; increased synovial fluid alpha-defensin (positive immunoassay or lateral-flow as-say); positivity of ≥2 samples with the same pathogen; >50 CFU/mL of any organism after sonication; and the presence of ≥ 5 neutrophils in ≥5 one high-power field HPF or occurrence of visible microorganisms in the histology. The infection diagnosis was defined as likely in the presence of two of the following criteria: signs of prosthesis loosening on radiological imaging; previous difficulty in wound healing; a recent history of fever; periprosthetic purulence; C-reactive protein (PCR) > 1 mg/dL; >1500/μL leukocytes or >65% PMN in the synovial fluid; aspiration fluid resulting in positive microbiological cultures; intraoperative fluid or tissue resulting in a single positive microbiological culture; >1 CFU/mL of any organism after sonication; occurrence of ≥5 neutrophils in one high-power field (400× magnification); and positivity of WBC scintigraphy.

Alongside this review, in order to enhance the success of a proper diagnosis, classic serological markers, including ESR, PCR, and Cell Blood Count (CBC), are widely used [30] as they are easy to obtain, cost-effective, and provide surgeons with rapid determination from a single blood test [19].

Consequently, many articles have researched the utility of laboratory tests, culture, and pathology in diagnosing PJI prior to two-stage revision [31,32,33]. The traditional inflammatory markers (VES and PCR) are highly effective for predicting PJI before revision arthroplasty [32], but they may not be suitable for the decision to replant. Among traditional inflammatory blood biomarkers, preoperative serum PCR has been considered the most reliable for identifying PJI [34], but it has never been associated with the utility to predict the success of PJI surgery.

The NLR, MLR, and PLR are readily available parameters that might have discriminative power regarding the outcome in arthroplasty [35].

Specifically, the NLR, PLR, MLR, and SII are the most common inflammatory blood factors studied [24]. Neutrophils, cells that belong to the innate immune system, constitute the main and most frequent parameter of expression of the inflammatory state. On the other hand, circulating monocytes, through their activation into macrophages, are specifically involved in the reparative phenomena of cellular or tissue damage. Lastly, lymphocytes perform their regulatory function of the body’s immune activity through the release of pro-inflammatory cytokines and cytolysis activity. The relationships defined between the different classes of cells and their ratios, that are inversely modified in inflammatory states, may represent potential and useful indicators suitable for directing the diagnosis of PJI [36]. Among these, the SIRI, a parameter obtained from the ratio of neutrophil, monocyte, and peripheral lymphocyte counts, could thus be considered a robust index of inflammatory status and it can therefore be applied to multiple settings [37], including that of prosthetic infections. It is worth considering that recently published scientific papers have emphasized the current scarcity of additional diagnostic tools that would allow for the early diagnosis of PJI alongside proven techniques clearly defined in current clinical practice (in particular molecular methods, imaging tests, microbiological and histological analysis, and the assessment of biomarkers in serum/synovial fluid samples) [38,39,40,41]. The search for an actual and cheap inflammatory marker to be used to improve monitoring early inflammation/infection after arthroplasty is still ongoing.

Many studies in the literature have described a fair diagnostic accuracy of the MLR, NLR, and PLR value in PJIs [33]. In particular, Tirumala and Klemt declared that the PLR, together with the platelet/mean platelet volume ratio (PVR), can be used in association with hematologic and aspirate markers to increase the accuracy of PJI diagnosis in total joint arthroplasty [42,43] and in total hip arthroplasty [44]. On the other hand, according to Ecki et al. [39], PCR and NLR values, either individually or in combination, before the second-stage revision arthroplasty were not predictive of either infection recurrence or cure within two years of follow-up.

As far as is known, no study has detected the role of the SIRI, SII, and AISI in the diagnosis of PJI and in defining the most appropriate time to ensure the success of the revision. Moreover, no study has ever defined the role of such biomarkers in identifying the most appropriate time for reprosthesing.

As a result, the identification of biomarkers to select patients likely to benefit from replant is a clinically unmet need and a critical issue of clinical research.

Our retrospective observational study focused on the prognostic value of six different cell ratios by evaluating their trend from explantation to replantation in a heterogeneous cohort of patients undergoing two-stage revision for chronic PJI.

Among the forty-two patients enrolled who were explanted with intent to replant, nineteen underwent a complete two-stage revision with a successful response.

Based on our findings, all the patients with successful revision had higher values of the SIRI (3.08 DS 1.7 and *p*-value 0.04) and MLR (0.4 DS 0.2 and *p*-value 0.02) at the time of EP, with a significant relationship between these parameters at explantation (and spacer insertion) and their values before replanting.

Additionally, according to our results, higher values of the delta-SIRI (−2.3 DS 1.8 and *p*-value 0.03) and delta-PCR (−46 DS 35.7 and *p*-value 0.03) were associated with a favorable outcome.

Examining the results collected, the values of the SIRI and PCR at 4 weeks from the EP were higher in patients who showed a successful result at reimplantation and during the follow-up.

On the other hand, with the measurements performed in the present study, no correlation was found between the other parameters before the treatment and at the replantation moment.

The SIRI and SII represent novel composite indices integrating three independent white blood cell subsets and platelets and they are considered reminiscent of the interaction of thrombocytosis, inflammation, and immunity. In most of the studies conducted so far, the SII and SIRI were found to be powerful prognostic indicators of outcome in different kinds of tumors, representing a promising tool for treatment strategy decisions [35,45,46]. They were also associated with cardiovascular death and all-cause death [47].

As the current standard treatment of chronic PJI consists of two-stage revision (removal of the implant infected and spacer implantation, antibiotic therapy, and implant of the new TJA), it could be useful to define the possibility of predicting the best moment to reimplant.

According to our results, it might be possible to predict the trend of a two-stage revision by studying the changes in the inflammatory factors SIRI and MLR. Considering the speed, easiness, and cheapness of obtaining these parameters from the mere CBC, these new biomarkers could reach everyday clinical practice.

The present study has some limitations. First, it includes a very small sample size and a relatively heterogeneous population. It must be considered that, despite the increasing numbers of patients affected by PJI, these are not common conditions. Moreover, there are still no meticulous guidelines or diagnostic algorithms going into depth on therapeutic strategies on this topic. Therefore, the results of the present study are of utmost importance, because they are supposed to encourage surgeons and clinicians to establish the best moment to execute the reimplantation.

More extensive studies analyzing a larger cohort of patients undergoing PJI are required to validate the promising results of this study.

## 5. Conclusions

The system inflammation response index (SIRI) is a novel composite index now considered a powerful and reliable indicator to reflect inflammatory conditions.

In particular, according to this study, it may represent an inflammatory marker whose trend could be useful in the prediction of success in two-stage revision for PJI and could help in predicting the clinical course of the response to the PJI. The data collected from this study hypothesized that the SIRI combined with ratio markers (MLR) may increase the accuracy of establishing the best moment to replant. These blood tests provide surgeons with important information on inflammatory–infectious status due to their simplicity, accessibility, and short waiting time.

In conclusion, our study seems to suggest that, in patients with PJI undergoing EP, the SIRI and MLR values and delta-SIRI and delta-PCR values are predictive of a favorable surgical outcome. These indices could guide clinicians in establishing the timing of reimplantation, but further studies are needed.

## Figures and Tables

**Figure 1 healthcare-12-00867-f001:**
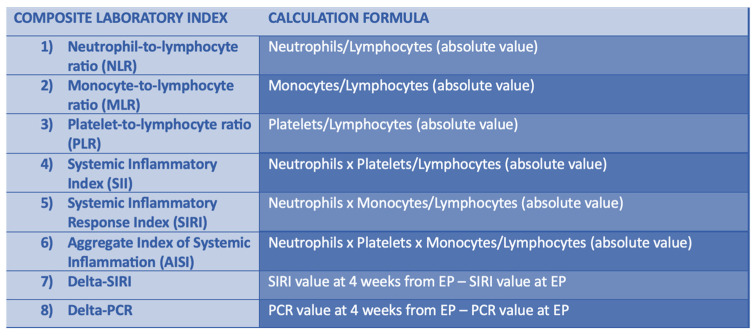
Composite laboratory index and their formula.

**Figure 2 healthcare-12-00867-f002:**
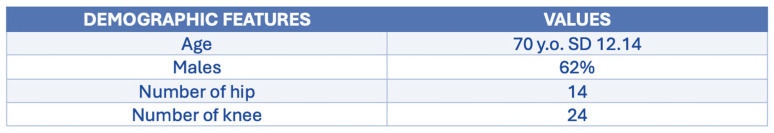
Demographic features.

**Figure 3 healthcare-12-00867-f003:**
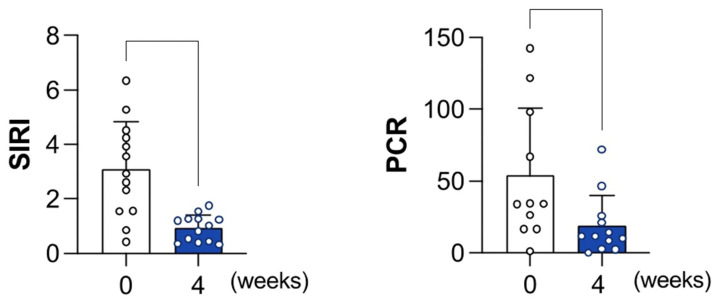
The SIRI and PCR trends at 4 weeks from the EP.

## Data Availability

Data are contained within the article.

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
