# Peer review of "Systemic Inflammation Response Index (SIRI) and Monocyte-to-Lymphocyte Ratio (MLR) Are Predictors of Good Outcomes in Surgical Treatment of Periprosthetic Joint Infections of Lower Limbs: A Single-Center Retrospective Analysis"

_healthcare, 2024, doi:10.3390/healthcare12090867_

Round 1

Reviewer 1 Report

Comments and Suggestions for Authors

The authors have investigated an interesting topic aimed at improving patient outcomes after PJI. They examined the prognostic value of several different blood markers and ratios in patients diagnosed with chronic PJI planned for a two-stage approach, in a center specialized in PJI treatment.

Abstract: The introduction section is overly long and lacks clarity. It would be more effective to briefly state the background and clearly outline the objectives.

Line 40: The statement "the other twenty-three patients did not replant due to persistent infection" lacks clarity. Did they undergo conversion to a 1.5 stage approach or receive no treatment at all? How was the persistent infection managed in these patients? Specific treatment should be indicated.

Line 42: Phrases like "higher probability of infection resolution" and "favorable outcomes" lack clarity regarding the outcomes being referred to—whether it pertains solely to infection resolution or includes other outcomes.

Line 45: Starting the conclusion with "Our study seems to suggest that…" introduces uncertainties regarding the study results. It would be better to reformulate this phrase.

Line 46: "in patients with PJI undergoing EP" - Where is the definition of EP in the manuscript? Is it referring to explantation?

Lines 32-34: Laboratory parameters (PCR, NLR, MLR, PLR, SIRI, SII, and AISI) were evaluated at explantation, at 4, 6, and 8 weeks after surgery, and at reimplantation. Often, evaluation is also done around 10-14 days post-surgery. Were any laboratory parameters evaluated during this timeframe or between day 0 to 30 after explantation? If not, why?

Lines 140-147: References 27 and 44 seem to address the same definition of chronic PJI. It should be clarified or only one reference should be used to inform readers about the chosen inclusion criteria.

The introduction cites the criteria defined by the EBJIS, but they are not applied in the study methods. This discrepancy can confuse readers about which criteria or definition to follow. If the EBJIS criteria are more complex and based mostly on laboratory parameters, while the IDSA criteria are more easily applicable in clinical practice, it might be more useful to include patients selected with the EBJIS definition and then investigate the laboratory ratios proposed by the study.

The results section is unclear and difficult to understand. Essentially, two groups were compared: 19 patients reimplanted with a two-stage approach and 23 patients who underwent EP with a planned two-stage approach but couldn't proceed due to persistent infection. What happened to the 23 patients? Did they wait longer for a second stage? Were the types of infection and antibiotics in the different groups considered, as they could influence the results? Was there a statistical analysis of demographic factors, risk factors, and comorbidities among patients who successfully completed the two-stage procedure versus those with persistent infection and DIAR?

In Figure 1, what does "one tibia" represent? Is it the knee? Are only periprosthetic infections considered, or does it include infections after osteosynthesis?

If the cutoff for chronic PJI was >4 weeks, were there patients with varying durations of infection? Considering patients with different durations (e.g., 5 weeks vs. 2 years) could be important. How can the results be applied in clinical practice to influence antibiotic duration after explantation? Could they help in preoperatively deciding whether to convert a planned two-stage approach to a one-stage revision?

Comments on the Quality of English Language

Fine.

Author Response

  • Abstract: we re-arranged the version trying to make it shorter and more incisive.
  • Line 40: Of the 23 patients who did not undergo prosthesis reimplantation, some (the most) retained the spacer in place, thus falling under the approach that is termed 1.5 stage approach; others (fewer in number) were at a later stage submitted to resection arthroplasty and arthrodesis (Girdlestone technique) or chronic suppressive antibiotic therapy because of increased risk due to comorbidities; finally, a few were over time lost to follow-up.
  • Line 42: “higher probability of infection resolution” refers to absence of signs of infection at the last available follow up; “good outcome” refers primarily to the absence of signs of inflammation impacting functional recovery at the last outpatient evaluation.
  • Line 45: The results of our study suggest that, in patients with PJI undergoing EP, SIRI and MLR values and delta-SIRI and delta-PCR values could be predictive of favorable outcome.
  • Linea 46: “EP” acronym inserted in line 32.
  • Line 32-34: The choice of parameters evaluated at 10-14 days or between day 0 to 30 after surgery was taken considering the occurrence of complications reported during the explantation procedure: (e.g., in case of massive blood loss intraoperatively, it was necessary to repeat the determination of CBC); rarely in this timeframe PCR was re-determined, to do not influence the result and create bias.
  • Line 140-147: “Chronic PJI” was defined as the infection that, regardless of origin, presented in a delayed form (≥4 weeks after surgery (usually 3 months to 3 years)) associated to at least 3 weeks of symptom duration.
  • We agree with Your consideration. In order to simplify without generating discrepancy on with definition was used, we adopted for the latest criteria for the diagnosis of PJI published in 2021 the European Bone and Joint Infection Society (EBJIS).
  • At the last available follow-up (average 11 months), the 23 patients who couldn't proceed to reimplantation due to persistent infection were directed to different diagnostic-therapeutic process (spacer retention/joint resection with chronic suppressive antibiotic therapy) or were lost to follow up. Our database included patients’ demographic factors, risk factors, and comorbidities but they were not submitted to analysis. Also, the different types of infection among the two groups were not taken into account, but it should be employed in future analysis.
  • Figure 1: we agree with this point, it was unclear (so we decide to remove it on the re-submitted file). It referred to a single case of left knee megaprosthesis (MUTARS) for giant-cell neoplasm of proximal tibia, twice submitted to two-stage revision, but at the end fallen into amputation because of concomitant diagnosis of an extensive pseudoarthrosis of tibia. It anyway belongs to “knee group”.
  • The enrolled patients had been diagnosed with prosthetic infection at very different times, particularly since our institution constitutes a very large catchment area of patients from other hospital hubs as well. In fact, comparing infections with different durations can be useful because each one tends to manifest a different individual course: during this progression, the level of inflammation, which then affects the parameters we assessed, changes, thus resulting in a different trend to be analyzed.
  • Our results may demonstrate that patients who achieve higher SIRI and PCR values in the early phases from intervention are more likely to achieve a complete or nearly complete success, thus predicting a better outcome.
  • Regarding the possibility of exploiting indices in defining the duration of antibiotic therapy, this might be feasible if coefficients of sensitivity and specificity, as well as precise cut-offs, would be identified in further studies, thus guiding therapeutic decisions to discontinue or continue antibiotic therapy.
  • Secondly, in this phase of our study, it is difficult to predict the applicability of the potential use of the evaluated markers in the one-stage revision, in particular considering it is applicable in selected cases and specialized centers. Specifically, considering that the contraindications to one-stage revision make the possibility of applying this procedure very restrictive, it is difficult to consider the margins of adaptation to this context. Above all, it is known that the currently most impactful contraindications for one-stage revision are infection caused by a drug-resistant bacteria, non-availability of appropriate antibiotics and presence of a sinus tract, since it is difficult to apply to the most frequent PJIs we are dealing with nowadays.

Reviewer 2 Report

Comments and Suggestions for Authors

This is a very informative article well written by the authors.

But a few major concerns are highlighted here.

1.Nineteen patients completed the two-stage revision process. Among them, none showed signs of re-infection or persistence of infection at the last available follow-up.

Kindly mention the duration of the follow-up.

Is long-term follow-up being done for the above group of patients, and are they on regular follow-ups?

2. Kindly provide the epidemiological data of the cohort included in the study with particular reference to

a. associated comorbidities 

b. underlying medications

c. any underlying dermatological  conditions  

3. Please provide the SENSITIVITY and SPECIFICITY values of  

Systemic inflammation response index (SIRI) and monocyte-to-lymphocyte ratio (MLR) as they are  critical to assessing its usefulness in predicting outcomes in PJI.

4. In the current study SIRI and MLR variables were used in a small cohort size. Whether the outcomes derived from the above variables concerning PJI shall be statistically significant and applicable to large groups?

Thank you and we look forward to hearing from you soon.

Comments on the Quality of English Language

Good quality english

Author Response

  • The minimum follow up was 1 year and is now mentioned in the article (178).
  • Epidemiological data of the patients available were included in the study. No dermatological conditions nor underlying medications were available in our database.
  • Sensivity and specificity of SIRI and MLR was not mentioned in this study because the small number of patients the data were not statistically significative, but a study with larger number of patients should obviously focus on these parameters.
  • We are confident that the values of SIRI and MLR applied to larger datasets yield statistically significant results. We have already begun collecting data for other studies, and the preliminary results appear to be positive, but it is crucial to standardize blood tests, timing, and procedures to obtain reliable data.

Reviewer 3 Report

Comments and Suggestions for Authors

Generally

If the work concerns the lower limb, why was elbow arthroplasty added to the group? Why do the authors in the results only point to the SIRI and CRP results? It is worth paying attention to the technical side of the table

Did the authors analyze what could be the cause of inflammation? It may be worth considering whether the surgical approach has an impact and the post-operative process, such as rehabilitation, which may affect inflammatory markers.

There is no information about the patients' inflammatory diseases in the last 3 months or additional injuries or increased stress hormones, which can cause long-lasting inflammation, or finally the patients' wishes, e.g. diet. What about allergies, have patients been tested for this? As you can see, the causes of inflammation are several and difficult to clearly assess and include criteria.

Most of the discussion is a combination of the introduction and the result, there are no references to other research conducted.

Methodology

Line 118 explains the PROCESS abbreviation

Figs 1 and 2 seem unnecessary due to the description in the text.

Other results are missing, such as NLR, PLR, AISI and SII

Discussion

Line 206-222 Should be moved to Introduction. As an explanation, the use of SIRI and MLR

Line 223 if many, why is there one footnote? This is not a meta-analysis. Additionally, this paragraph lacks a reference to the authors' study. Without this, this paragraph loses its meaning and should be included in the introduction.

Author Response

  • “Elbow” it was a misprint from a previous version of the document, which has been therefore removed with simultaneous modification of the corresponding table.
  • We only pointed to SIRI and CRP in the results because the other studied parameters were not statistically significant (added on the text).
  • The possible causes of inflammation to be considered in patients referring to our center, who are usually very complex and multi-comorbid, are countless. Certainly, in future studies it might be useful to select the most relevant ones, also according to the literature available to date. Among these we believe that the most impactful are other concomitant inflammatory diseases and rehabilitation recovery. Regarding surgical strategy, our center tries to adopt protocols as standardized as possible, except for intra-operative complications or other patient-dependent issues.
  • Concerning allergies, unless there are known drug allergies specifically among antibiotics, it is not our practice to request a specialist evaluation. Instead, in case of documented or reported allergic reaction (particularly type 1) we proceed to adopt a regimen that does not imply problems or cross-reactions and at the same time we request allergy counseling to test available drug alternatives.
  • Line 118: we explained PROCESS
  • We considered that expressing patients’ demographics features and involved sites of infection through figures was more immediate and suitable for the reader, although we could surely have elicited them in the text.
  • We only pointed to SIRI and CRP in the results because the other studied parameters were not statistically significant (added on the text), as their trend did not demonstrate a statistically significant correlation with the outcome of prosthesis explantation or with the two-stage revision final success.
  • Line 206-222: we decided not to move it to the introduction, but we tried to improve the explanation.
  • Line 223: we implemented footnotes, and we added an indirect reference to the authors' study in filling the gap underlined.

Round 2

Reviewer 1 Report

Comments and Suggestions for Authors

Thanks for the answers.

Reviewer 2 Report

Comments and Suggestions for Authors

Thank you for your critical comments and feedback. 

Comments on the Quality of English Language

Quality is good.